Daily accumulation rates of floating debris and attached biota on continental and oceanic island shores in the SE Pacific: testing predictions based on global models

Rech Sabine sabinerech01@gmail.com 1 2
Arias Rene Matias 1
Vadell Simón 1
Gordon Dennis 3
Thiel Martin 1 2 4
1 Departamento de Biologia Marina, Facultad de Ciencias del Mar, Universidad Catolica del Norte , Coquimbo , Chile
2 Center for Ecology and Sustainable Management of Oceanic Islands ESMOI, Universidad Catolica del Norte , Coquimbo , Chile
3 National Institute of Water and Atmospheric Research (NIWA) , Kilbirnie , Wellington , New Zealand
4 Centro de Estudios Avanzados en Zonas Áridas (CEAZA) , Coquimbo , Chile
Mahmood Haider
Electronic publication date: 2023 Jul 27
Publication date: 2023
Volume: 11
Electronic Location ID: e15550
Received 2023 Jan 17; Accepted 2023 May 23
Copyright: ©2023 Rech et al.
Copyright year: 2023
Copyright holder: Rech et al.
License: This is an open access article distributed under the terms of the Creative Commons Attribution License, which permits unrestricted use, distribution, reproduction and adaptation in any medium and for any purpose provided that it is properly attributed. For attribution, the original author(s), title, publication source (PeerJ) and either DOI or URL of the article must be cited.
License URL: https://creativecommons.org/licenses/by/4.0/

Keywords: Anthropogenic marine debris, Floating litter, Marine currents, Marine invertebrates, Plastic pollution, South Pacific Subtropical Gyre, Marine debris accumulation patterns, Invasive species, Dispersal by rafting, Rapa Nui (Easter Island)

Funding: ANID (Agencia Nacional de Investigación y Desarrollo, Chile) FONDECYT POSTDOCTORADO 2020 3201074 This work was supported by ANID (Agencia Nacional de Investigación y Desarrollo, Chile) in the program FONDECYT POSTDOCTORADO 2020, project No. 3201074. The funders had no role in study design, data collection and analysis, decision to publish, or preparation of the manuscript.

==============================
Background

Long-distance rafting on anthropogenic marine debris (AMD) is thought to have a significant impact on global marine biogeography and the dispersal of non-indigenous species. Therefore, early identification of arrival sites of AMD and its epibionts is crucial for the prioritization of preventive measures. As accumulation patterns along global coastlines are largely unstudied, we tested if existing oceanographic models and knowledge about upstream sources of litter and epibionts can be used as a simple and cost-efficient approach for predicting probable arrival sites of AMD-rafting biota in coastal zones.

Methods

Using the Southeast Pacific as a model system, we studied daily accumulation rates, composition, and minimum floating times of AMD with and without epibionts on seven sandy beaches, covering the oceanic environment (Rapa Nui/Easter Island) and three regions (south, centre, north) along the Chilean continental coast, over a minimum of 10 consecutive days, and we contrast our results with predictions from published models.

Results

Total AMD accumulation rates varied from 56 ± 36 (mean ± standard deviation) to 388 ± 433 items km−1 d−1 and differed strongly between regions, in accordance with local geomorphology and socioeconomic conditions (presence of larger cities and rivers upstream, main economic activities, etc.). Daily accumulation of items with pelagic epibionts (indicators of a pelagic trajectory) ranged from 46 ± 29 (Rapa Nui) to 0.0 items km−1 d−1 (northern continental region). Minimum floating times of rafts, as estimated from the size of pelagic epibionts, were longest in the South Pacific Subtropical Gyre’s (SPSG) centre region, followed by the high-latitude continental region under the influence of the onshore West Wind Drift, and decreased along the continental alongshore upwelling current, towards lower latitudes. Apart from pelagic rafters, a wide range of benthic epibionts, including invasive and cryptogenic species, was found on rafts at the continental beaches. Similarly, we present another record of local benthic corals Pocillopora sp., on Rapa Nui rafts.

Discussion

Our results agree with the predictions made by recent models based on the prevailing wind and surface current regimes, with high frequencies of long-distance rafting in the oceanic SPSG centre and very low frequencies along the continental coast. These findings confirm the suitability of such models in predicting arrival hotspots of AMD and rafting species. Moreover, storm surges as well as site-related factors seem to influence AMD arrival patterns along the Chilean continental coast and might cause the observed high variability between sampling sites and days. Our results highlight the possible importance of rafting as a vector of along-shore dispersal and range expansions along the SE Pacific continental coast and add to the discussion about its role in benthic species dispersal between South Pacific oceanic islands.

Introduction

Rafting on anthropogenic marine debris

Anthropogenic marine debris (AMD), consisting primarily of plastics, is a global problem with a wide range of impacts (Bergmann, Gutow & Klages, 2015; Napper & Thompson, 2020). The transport and dispersal of marine biota on such items (i.e. rafting) is now known to be a common process, both in nearshore and offshore environments (Kiessling, Gutow & Thiel, 2015; Rech et al., 2018; Rech et al., 2021; Haram et al., 2021; Haram et al., 2023), which is likely to increase with increasing availability of such objects. Apart from the general importance of AMD rafting for marine species dispersal and biogeography, there is growing evidence and concern about its role in the dispersal of invasive and potentially harmful species (Ivkić et al., 2019; Simkanin et al., 2019; Faria & Kitahara, 2020; García-Gómez, Garrigós & Garrigós, 2021). Such transport can occur over long, even trans-oceanic, distances (e.g., Hoeksema, Roos & Cadée, 2012; Hoeksema, Roos & Cadée, 2015; Hoeksema, Pedoja & Poprawski, 2018; Holmes et al., 2015) and can affect remote and susceptible environments. The largest documented rafting event of marine invertebrates on artificial structures was the transport of almost 300 coastal Japanese taxa across the North Pacific on items detached by the Tōhoku tsunami in Japan in 2011 (Carlton et al., 2017). However, most rafting events probably remain undetected, due to lack of the local monitoring required for early detection of non-indigenous species (NIS) on rafts in a particular region. Unfortunately, due to limitations in time, funding, and access to many coastal sites, regular monitoring is not feasible on most global coasts. Therefore, the ability to predict and identify susceptible arrival sites with simple and efficient measures is crucial for prioritisation of valuable monitoring and protection measures.

Distribution of AMD and epibionts on global and regional scales

Modelling efforts and observational studies provide a broad overall picture of sources, pathways, and sinks of floating marine litter on a global scale: most floating litter enters the marine environment from terrestrial (coastal or inland) sources and is then dispersed by winds and surface currents (e.g., Lebreton, Greer & Borrero, 2012; Chassignet, Xu & Zavala-Romero, 2021; Chenillat et al., 2021). An estimated 70% of land-sourced litter items strands along coastlines, while the rest accumulates in the open sea (Chenillat et al., 2021; Onink et al., 2021). On their way towards the open ocean, smaller items easily lose buoyancy and sink to the seafloor (Ryan, 2015; Fazey & Ryan, 2016), while larger and stable items can float for long periods (several years) and accumulate in the main oceanic gyres (e.g., Eriksen et al., 2014). These persistent long-distance rafts are the most important ocean-travelling objects that may enhance the dispersal of attached benthic (coastal) epibionts, from bacteria and algae to larger invertebrates, on interregional and trans-oceanic scales (e.g., Carlton et al., 2017). Along continental coastlines, accumulation frequencies of AMD and attached epibionts from long-distance sources are determined primarily by the direction of the prevailing surface currents (Thiel & Gutow, 2005a; Morales-Caselles et al., 2021), including up- and downwelling effects that push floating items in an offshore or onshore direction, respectively (e.g., Pereiro, Souto & Gago, 2019; Van Sebille et al., 2020). For rafts from nearby coastal sources, upstream population density and, in some regions, presence of rivers can be important predictors as well (Chenillat et al., 2021). To date there is no clear pattern of the distribution of floating litter or the underlying mechanisms, but large regional variation is predicted along global coastlines (Chenillat et al., 2021). For example, long-distance dispersal is known to be mediated by the North Atlantic Gulf Stream and North Atlantic Drift, which frequently carry floating rafts and their epibionts (both benthic and pelagic) from western to eastern Atlantic waters and shores (Hoeksema, Roos & Cadée, 2012; Hoeksema, Roos & Cadée, 2015; Hoeksema, Pedoja & Poprawski, 2018; Holmes et al., 2015). Similarly, in the Southern Hemisphere, the circumpolar West Wind Drift (WWD) hits the southern Chilean continental coast and has a known role in past and present dispersal of marine rafting taxa on floating items, particularly kelp (e.g., Foighil et al., 1999; Waters, 2008; Nikula et al., 2010).

Within a coastal region, deposition of floating debris on beaches depends on a variety of factors related to climatic conditions and local geomorphology, amongst others (e.g., Critchell et al., 2015; Critchell & Lambrechts, 2016; Ourmieres et al., 2018), and a heterogeneous distribution of long-distance floating litter along shorelines seems highly likely. Nevertheless, while there are many examples from relatively short coastlines, there are no comparable studies along larger coastal stretches. Moreover, sampling and classification systems as well as temporal resolution between existing studies are not standardized (Browne et al., 2015). For a reliable comparison of litter accumulations, regular samplings with short sampling intervals (e.g., daily samplings, ensuring brief residence times of litter on the shore) are needed (Smith & Markic, 2013; Ryan et al., 2014; Browne et al., 2015).

The eastern South Pacific Ocean as a model system

The eastern South Pacific Ocean is an ideal model system to test globally valid hypotheses about distribution and accumulation rates of AMD and its epibionts along continental coastlines and in the open ocean. The centre of the South Pacific Subtropical Gyre (SPSG), with the oceanic island Rapa Nui (Easter Island), is located about 4,000 km off the Chilean coastline, which allows for direct comparisons between oceanic (island) beaches and continental beaches. Moreover, the Chilean coastline covers latitudinal ranges from 17°S to 56°S, with a relatively clear biogeographic zonation (Camus, 2001; Thiel et al., 2007), which allows for inter-regional comparisons. The counterclockwise currents of the SPSG transport floating matter from both eastern (e.g., Van Gennip et al., 2019) and western shores (e.g., Australia, Western Oceania) and even the Indian Ocean (Maes et al., 2018; Chenillat et al., 2021) to the gyre’s centre region. The southern part of the Chilean continental coast is subject to one of the major onshore current systems of the world, as it receives floating items and epibionts from the Western Pacific via the circumpolar WWD (Foighil et al., 1999; Waters, 2008; Nikula et al., 2010).

Alongshore, the coastal Humboldt current flows northwards, but a strong upwelling regime with offshore Ekman transport in the central and northern regions diverts floating items offshore (Marín, Delgado & Luna-Jorquera, 2003; Marín & Delgado, 2007; Thiel et al., 2007; Van Gennip et al., 2019). At latitudes between 40°S and 20°S, upwelling is subject to seasonal and sub-seasonal variability (Rutllant, Rosenbluth & Hormazabal, 2004; Thiel et al., 2007; Aguirre et al., 2021). Global models predict low accumulation frequencies and an almost exclusive SE Pacific origin of floating litter along the entire Chilean coast (Chassignet, Xu & Zavala-Romero, 2021; Chenillat et al., 2021; Onink et al., 2021). A recent model by Chenillat et al. (2021) restricts the predicted zone of arrivals of long-distance rafts to latitudes from 30°S to 47°S, with main accumulations between 40°S and 45°S, where the WWD reaches the South American continent, and the main transport direction is onshore.

In the SPSG centre region, floating and stranded AMD is traced primarily back to South Pacific high-sea fisheries and the Chilean coastline (Kiessling et al., 2017; Luna-Jorquera et al., 2019; Van Gennip et al., 2019; Thiel et al., 2021), from where floating items may arrive in less than two years (Van Gennip et al., 2019). However, most AMD rafters found on Rapa Nui beaches and in central SPSG waters are pelagic, and the supply of benthic rafters from the continental coasts is thought to be limited by a strong nutrient gradient between the highly productive coastal and the hyper-oligotrophic pelagic environment (Rech et al., 2018; Rech et al., 2021). Nevertheless, given the high frequency of floating litter arrivals in the region (Van Gennip et al., 2019; Thiel et al., 2021) and the scarcity of coastal monitoring, arrivals of continental benthic species cannot be entirely excluded and could easily go unnoticed.

Considering all this information, we propose that the accumulation frequencies of AMD with a pelagic trajectory and long floating times (as indicated by the presence and size of pelagic epibionts) depend primarily on the prevailing currents, with highest arrival frequencies in oceanic gyres (here: the Rapa Nui region) and in continental regions under the influence of prevailing onshore currents (here: the southern continental region). For regions outside of their influence or where offshore transport is prevalent, we hypothesize that rafts and epibionts are mostly locally sourced, with their frequencies depending on short-distance sources of litter and suitable rafting taxa. Furthermore, we expect AMD composition to reflect the main litter sources (local vs. distant) at each site, with highly buoyant and durable plastics dominating at the principal arrival sites (beaches in the SPSG centre region) and higher shares of local and short-lived items along the continental coast. To test these hypotheses, the present work compares accumulation frequencies as well as the composition of AMD and attached biota between oceanic (Rapa Nui) and southern, central, and northern Chilean continental shores.

As an overall aim, and given the lack of knowledge about AMD accumulation patterns along most global coastlines, the present study tests if high-risk arrival areas that may receive non-indigenous AMD rafters can be reliably predicted based on existing oceanographic models and knowledge of local/regional oceanographic conditions and upstream sources of possible rafting organisms. We propose that this can be used as a simple and efficient approach for the prioritization of cost- and time-intensive actions such as surveying and protection measures.

Material & Methods

Sampling regions and sites

Stranded AMD was sampled on seven sandy beaches: two beaches on Rapa Nui (Easter Island) in the centre of the South Pacific Subtropical Gyre (SPSG) and five beaches in three regions (south, centre, and north; 42°S to 29°S) along the Chilean continental coast, between early March and early May of 2021 (Fig. 1; Table 1). Along the continental coast, beaches with a known history of pelagic flotsam arrivals (i.e., with pelagic epibionts Lepas spp. and/or Jellyella sp.; see López et al., 2017; Rech et al., 2018), were chosen within the latitudinal range of floating litter arrivals, as predicted by Chenillat et al. (2021). To ensure comparability among all sampled beaches and avoid biases in litter quantification (see Browne et al., 2015), our study was limited to sandy beaches. Apart from these common features and to ensure broad validity of results, sampled regions and sites covered diverse biogeographic, oceanographic, and socio-economic characteristics of adjacent human settlements.

Figure 1 Sampling sites (red marks) in the South Pacific Subtropical Gyre (SPSG) and the South American continental coast.

(1) Beaches Anakena and Ovahe on Rapa Nui island in the SPSG center. (2) Beaches Mar Brava North and Mar Brava South on Chiloé Island (southern continental Chile). (3) Beaches Ritoque and Maitencillo in the central continental region. (4) Choros beach in the northern continental region. Streamlines show 1992–2021 mean 10-meter wind calculated from the ECMWF product. Human settlements (dark grey) and rivers (blue) are shown. WWD, West Wind Drift. HC, Humboldt Current.

Table 1 Sampled beaches with abbreviations, geographic position, section length, and start date of the sampling period.

Region	Beach	Code	Latitude	Longitude	Section length [m]	Start date	
South Pacific Subtropical Gyre: Rapa Nui	Anakena	Ana	−27.073	−109.323	250	04.03.2021	
Ovahe	Ova	−27.073	−109.314	80	04.03.2021	
Continental: South	Mar Brava North	MBN	−41.866	−74.009	1000	07.05.2021	
Mar Brava South	MBS	−41.912	−73.996	1000	07.05.2021	
Continental: Centre	Maitencillo	Mait	−32.662	−71.441	1000*1	21.04.2021	
Ritoque	Rit	−32.830	−71.523	1000*1	21.04.2021	
Continental: North	Choros	Cho	−29.254	−71.437	1000*2	14.03.2021	
Notes.

*1 subsampled in smaller sections on days 7, 8 and 9.

*2 only 500 m sampled on days 11 and 12.

Oceanic Rapa Nui is a small (20 km long) and very remote island with a relatively low population density (23.1 inhabitants km−2; INE, 2017) and no major putative sources of local litter input (e.g., rivers, ports, industry), located in the SPSG centre region and known to receive large quantities of floating AMD (e.g., Rech et al., 2018). Usually a hotspot for international tourism, the island was completely cut off during the COVID-19 pandemic, when samplings were conducted. In the southern continental region, samplings were conducted on the relatively pristine and scarcely populated (18.3 inhabitants km−2; INE, 2017) Chiloé Island, situated about 40 km off the Chilean mainland. The island’s main economic activity is aquaculture, which is restricted to the protected mainland-facing inner coast. The outer coast, where the sampled beaches are located (Fig. 1), is hit by the Antarctic Circumpolar Current (West Wind Drift). The central continental region, around the political and economic capital Santiago, is densely populated, with a highly intervened and urbanized coastline, featuring the cities Valparaiso and Viña del Mar, several industrial centres, two international ports, and river mouths. Although tourism and outdoor activities were strongly reduced due to pandemic-related restrictions during the sampling period, beaches were frequently used by residents of the region’s seaside resorts. In contrast, the northern continental sampling region around Choros beach is sparsely populated and situated about 80 km north of the region’s major towns Coquimbo and La Serena. Beach tourism and public life were minimal due to pandemic-related restrictions during the samplings, limiting beach use to artisanal fishing. The central and northern regions are under the influence of the along-shore Humboldt Current and predominant upwelling regimes. For more information on sampling sites, please see additional text in the supplement (Supplemental Text). All field and laboratory work related to the project was approved by the ethics committee of Universidad Católica del Norte in Coquimbo (approval number CECUCN N° 07/2020).

Daily AMD samplings

First, all beaches were inspected, and the future sampling sections were selected and marked. On the short Rapa Nui beaches, this section stretched along the whole length of each beach (Anakena: 0.25 km, Ovahe: 0.08 km; see Table 1). On the long continental beaches, a 1-km stretch with a representative level of beach litter (not significantly more or less polluted than the rest of the beach, based on visual inspection) and no immediate vicinity to common point-sources of terrestrial beach litter (car parks, cafés, beach sport installations) was selected. This length was chosen to minimize effects of heterogenous litter distribution and stranding patterns, as well as edge effects with the adjacent (not cleaned) beach sections. To maintain consistent sampling accuracy among all beaches and sampling days, Choros beach was subsampled in several small transects on three consecutive sampling days due to intense and heterogenous deposition of seafloor litter and biota (Fig. S1), inhibiting a thorough sampling of the complete 1-km section. Similarly, at Ritoque and Maitencillo beaches, the section had to be shortened to 500 m on the last two sampling days.

On the day before the first sampling (Day 0), the whole intertidal area, ranging from the water line to the highest visible tide line (see schematic overview in Fig. S2) was manually cleared of all visible litter items ≥ 1.5 cm along the whole transect (+20 m to each side on continental beaches). Starting from the next day (Day 1), the recent intertidal section, ranging from the water line to the highest recent (i.e. from the last 24 h; see Fig. S2 and Text S1) tide line was sampled on 12 consecutive days (10 days on Rapa Nui beaches) in the morning by walking along all visible accumulation lines and picking up all litter within a range of 1 m to each side. Cigarette butts were excluded from the study due to their enormous quantities on some beaches and because, according to our current knowledge, they are also not relevant in the context of long-distance dispersal. On each sampling day, all other litter items with a size of ≥ 1.5 cm were removed from the sampling section. Each item was quickly inspected upon collection and items with and without obvious biofouling were stored separately. Items with mobile or loosely attached biota were stored in a separate bag. The same was done with heavily fouled items, where mobile organisms might be hidden, for example, in dense aggregations of algae or hydroids. Items that were too big or too heavy to be removed from the beach were photographed with a size reference in situ. If biofouling was present on such items, it was documented photographically, and a sample of each distinguishable taxon was taken. All other litter items were taken to the field laboratory for further processing. In addition to daily quantitative samplings, raft items were sampled opportunistically outside of the defined sampling section at sampling sites along the continental coast. These items were used for the detection of additional rafting taxa but were not considered in quantitative analyses of litter frequencies or composition.

Sample processing in the field laboratory

In the field laboratory, litter items with no obvious biofouling were cleaned of attached sand by carefully submerging them in a bucket with freshwater. The item was then thoroughly inspected to detect any previously invisible biofouling. All items were counted and sorted into categories (see section “litter categories” below). Items with biofouling were stored for further treatment, while items without any biofouling were disposed of. Items with immediately evident, larger biofouling were first photographed with the epibionts still attached. If live epibionts were present, the whole item or the detached epibionts were placed in seawater to take pictures for future identification and then stored in 70% ethanol for further processing. Fouled items and epibionts were labelled and taken to the marine ecology laboratory at the Faculty of Marine Science of Universidad Católica del Norte in Coquimbo for further processing.

Processing of fouled items and epibionts

In the laboratory, all fouled items and epibionts were photographed with a size reference, and attached biofouling was inspected with a dissecting microscope. Photos of epibionts were taken with the “microscope” mode of a camera (Olympus Tough TG-5, Olympus, Tokyo, Japan). Scanning electron microscope (SEM) images were taken of some encrusting bryozoan species and are available at figshare data repository (https://doi.org/10.6084/m9.figshare.23610873.v2). Preliminary identification of epibionts was based on identification guides, state-of-the-art literature, original descriptions and re-descriptions of species (e.g., Hayward & Ryland, 1995; Häussermann & Försterra, 2009). Final identification was based on the expertise of professional taxonomists (see acknowledgements), with nomenclature following the World Register of Marine Species (WoRMS Editorial Board, 2023). Habitat (pelagic/benthic) was identified for each species based on scientific literature. Only species known to exclusively occur on floating objects offshore (also termed “obligate rafters”, see for example Thiel & Gutow, 2005b; Carlton et al., 2017 supplement) were counted as “pelagic”, with only three taxa fulfilling this condition: The encrusting bryozoan Jellyella sp., the goose barnacles Lepas spp., and the nudibranch Fiona pinnata. All species known to inhabit the inter- or subtidal zones in any stage of their life cycle were categorized as “benthic”. The presence of strictly pelagic epibionts was used to identify rafts with an offshore trajectory (as opposed to rafts from local sources arriving via nearshore currents). If Lepas spp. were present on a raft, the capitulum length of the largest individual of each distinguishable species (Lepas sp., L. anatifera, L. pectinata, L. australis) was measured and used to calculate the item’s minimum floating time, based on published growth rates (see Thiel & Gutow, 2005b). This approach is commonly used in ecologic as well as forensic studies (e.g., Macaya et al., 2005; Rothäusler et al., 2011; Magni et al., 2015).

Determination of buoyancy

Following the removal of attached sand, each item was submerged in a bucket filled with seawater and its behaviour (floating/sinking) was observed. If an item did not show a clear sinking or floating behaviour when first tested, for example by staying in the mid-water zone, or if its behaviour seemed to be biased, for example due to the formation of small air bubbles that might keep the item floating, the assay was repeated. If an item’s buoyancy could not be established after several (three to four) repetitions, it was categorized as “buoyancy unclear”. Firmly attached epibionts (if present) were not removed from the items before the assays, as we wanted to test the actual buoyancy upon stranding. Buoyancy was tested for an extensive sample (almost 10,000 items) from central and southern continental beaches (Table S1). Based on the results obtained, the proportion of positively buoyant items was calculated for each litter category (Table S2).

Litter categories

The distinction of rafts that likely arrived floating from long-distance sources and those that arrived from “local” sources, either floating over a short distance or not floating at all (for example by direct littering at the beach), is a major challenge of the present work. Items suited for long-distance marine rafting are positively buoyant and highly durable items that can support long ocean travels and macroscopic biofouling without breakage or sinking, e.g., hard plastic fragments, bottle caps, crates or containers (e.g., Morales-Caselles et al., 2021; Rech et al., 2021). Following a conservative approach, only items fulfilling these conditions were regarded as potentially coming from distant sources and joined in the category “Hard plastics” (HPL; see Table 2). On the other hand, locally-sourced AMD usually contains higher percentages of non-durable items that break or sink easily, like plastic food wrappers or bags, as well as negatively buoyant items, like textiles or glass fragments (see also Morales-Caselles et al., 2021). To test our hypothesis that locally-sourced items would reflect the prevailing coastal activities or point-sources of litter input at each sampling site, all items of putative “local” origin (i.e. all items that did not meet the criteria for the HPL category, see above) were sorted into subcategories based on item type and material. Items or materials that were not common enough for a separate category were pooled. Buoyancy was calculated for each item category based on the floating experiments described in the previous section (see Table 2).

Table 2 Litter categories used in this study.

Characteristic traits (buoyancy and longevity) and description are shown for each category.

Category	Description	Buoyancy	Longevity	
Hard plastics (HPL)	Rigid plastic items and fragments, except for PET fragments.	Positive (97.5 ± 4.2%)	High	
Thin plastics (TPL)	Thin plastic items and fragments. E.g., food wrappers, bags and single-use dishes and cutlery.	Positive (93.6 ± 6.2%)	Low	
Ropes	Plastic ropes and rope bundles*a.	Medium (71.2 ± 32.4%)	Medium	
Other plastics (Other, PL)	All other plastic items, e.g., Foams, PET fragments, monofilament line, sanitary masks, and wet wipes.	Medium (70.8 ± 39.6%)	Mixed	
Other, Mix	Paper & cardboard, glass & ceramics, metal, rubber, textile & shoes, processed wood, food rests, unidentified materials.	Medium/low (61.0 ± 35.5%)	Mixed	
Notes.

a See photo in Fig. S6 for examples of rope bundles. Buoyancy = proportion of positively buoyant items in each category based on floating experiments (see Tables S1 and S2).

Data analysis & statistics

Daily AMD arrival frequencies were calculated for each beach and sampling day as the total number of items found in the sampled section, divided by the section’s length (see Table 1 for lengths of sampling sections). Regional differences in AMD arrival frequencies were tested using permutational ANOVA (PERMANOVA), with “region” as a fixed factor, based on a Euclidean distance matrix of the arrival frequencies (items km−1) per beach and sampling day (Table S3), performed with 9,999 permutations, under the condition of unrestricted permutation of raw data. For statistical analysis of litter composition, the contribution of each of the five main litter categories used in this study (Hard plastics, Thin plastics, Other plastics, Ropes, Other/Mix) to AMD found on the sampling transect during daily samplings was expressed in percentages (%). Again, regional differences were tested by PERMANOVA, based on an Euclidean distance matrix of the normalized litter composition for each beach and sampling day (see Supplemental File), under the conditions specified above. Moreover, the main litter categories defining AMD in each region (as well as differences between those regions) were calculated using similarity percentage (SIMPER) analysis, based on Euclidean distances, and graphically depicted through principal component analysis (PCA) ordination, based on a variance–covariance matrix. PERMANOVA and SIMPER analyses were done using PRIMER 6 + PERMANOVA software (Anderson, Gorley & Clarke, 2008). PCA was performed with PAST software (Hammer, Harper & Ryan, 2001). Species richness per raft and number of species per beach were calculated conservatively, considering the minimum number of distinguishable species.

Results

Daily accumulation rates

More than 11,000 litter items were sampled and analysed during the study. Total AMD accumulation rates were significantly lower (56 ± 36 items km−1 d−1) on the relatively pristine beaches of the scarcely populated southern continental region than in all other regions (PERMANOVA, p = 0.0004). In contrast, the densely populated central region, with frequent beach use and diverse point sources of terrestrial litter input (river mouths, ports, industry), held the highest frequencies (388 ± 433 km−1 d−1; Fig. 2A; Table S3). However, there were relatively large differences between sites of the same region as well as between sampling days (Fig. 2A and Fig. S3).

Figure 2 Daily AMD accumulation rates.

Above: all AMD items. Below: AMD items with pelagic epibionts only. Boxplot indicates mean value (x), median value (—), interquartile range (box) and outliers (°). Ana, Anakena; Ova, Ovahe; MBS, Mar Brava South; MBN, Mar Brava North; Rit, Ritoque; Mait, Maitencillo; Cho, Choros; SPSG, South Pacific Subtropical Gyre; RN, Rapa Nui; Conti, Continental coast. Statistically significant differences (PERMANOVA, p < 0.05) between regions are indicated by letters a, b, c.

Accumulation rates of items with pelagic epibionts differed strongly and significantly between sampled regions (PERMANOVA p = 0.0001; Fig. 2B). As hypothesized, they were significantly higher in the SPSG center region (Rapa Nui: 46 ± 29 items km−1 d−1) than in all continental regions. Along the continental coast, the central region received the highest frequencies of AMD with pelagic epibionts (4 ± 7 items km−1 d−1). In the southern region, arrival frequencies were much lower (0.3 ± 0.4 items km−1 d−1). However, as was the case for AMD in general, the deposition of items with pelagic epibionts was highly variable not only within each continental region, but also between sampling days (Fig. S4). This was particularly evident in the central continental region, where accumulation frequencies were exceptionally high on four sampling days on Ritoque beach (7 ± 9 items km−1 d−1), but not on Maitencillo beach (1 ± 1 items km−1 d−1; Table S3).

In the northern continental region, no items with pelagic epibionts were found during the 12-day sampling period on the 1-km sampling section of Choros beach. The relatively high deposition rates of AMD on this beach (Fig. 2A) were probably caused by strong winds, which deposited large quantities of subtidal (i.e. benthic) litter and biota on the section (see photo in Fig. S1). While Lepas spp. were the only pelagic species found on southern and central continental beaches, rafts from oceanic Rapa Nui beaches were mostly fouled by the pelagic bryozoan Jellyella sp.

Litter composition

The composition of litter washing ashore differed significantly between sampling regions (Fig. 3; PERMANOVA, p = 0.0001). The vast majority (70%) of AMD arriving on the oceanic Rapa Nui beaches were highly buoyant and stable hard plastic fragments and items (for buoyancy estimates, see Tables S1 and S2), washing ashore from the SPSG (Figs. 3 and 4 and Tables S4 and S5). In contrast, the contribution of such hard plastic items (HPL) from putative long-distance sources was much lower at all continental sites (Fig. 4, Tables S4 and S5). In line with our hypothesis, AMD composition in the sampled continental regions reflected the predominant local coastal activities: In the southern continental region, with no larger settlements upstream and where aquaculture is the only relevant economic activity, ropes were the most frequent and defining litter category (59% contribution, see SIMPER results in Table S5; Figs. 3 and 4). In the highly urbanized and industrialized central continental region, with frequent recreational beach use even during our samplings amid the COVID19 pandemic, AMD mainly comprised buoyant but non-durable thin plastic items like food wrappers (“TPL”, Figs. 3 and 4) from putative local sources, with a contribution of 67% to within-group similarity (see SIMPER, Table S5). The northern continental region, downstream of two larger cities and with artisanal fishing as its main economy, had the most diverse AMD composition (Fig. 4), with ropes and thin plastics contributing 36% and 32% to within-group similarity, followed by miscellaneous plastic items (“Other, PL”) contributing 23% (SIMPER, Table S5).

Figure 3 Result of PCA (principal component analysis) showing differences in litter composition between regions, and the principal litter categories causing these differences.

Oceanic_Rapa Nui: black circle = Ovahe, black square = Anakena; Continental_South: white circle = Mar Brava South; x = Mar Brava North; Continental_Center: white triangle = Ritoque, white square = Maitencillo; Continental_North: white rhomb = Choros; HPL = Hard plastics; TPL = Thin plastics.

Figure 4 Litter composition on sampled beaches.

“Total”, all litter items; “pel”, items with pelagic taxa only. MBS, Mar Brava South; MBN, Mar Brava North. SPSG, South Pacific Subtropical Gyre; RN, Rapa Nui. Conti, continental; S, South; C, Centre; N, North. HPL, Hard plastics; TPL, Thin plastics. For more specific information on item categories, please see Table 2. Numbers in parentheses = number of items.

Analysing litter with pelagic epibionts (i.e., indicating a pelagic trajectory) only, there are generally higher proportions of stable, hard plastic rafts (HPL), contributing 99% to within-group similarity in the oceanic Rapa Nui region (SIMPER, Table S5; Fig. 4). In the southern continental region, three categories (HPL, Ropes, Other/Mix) had equal contributions, but the low number of rafts with pelagic epibionts found during quantitative samplings in this region (n = 6) probably does not allow for robust results. Interestingly, in the central continental region, where a high number of rafts with pelagic epibionts were found (n = 76), non-durable thin plastic items (TPL) continued to be the most frequent (44% of all rafts, Fig. 4) and defining (60% contribution to within-group similarity, SIMPER, Table S5) category, indicating that even rafts carrying pelagic epibionts would probably have travelled for relatively short periods of time, i.e., over relatively short distances.

Minimum floating time based on epibiont size

Size of pelagic epibionts and estimated minimum floating times varied considerably between sampling sites. Maximum sizes of Lepas spp. were largest on AMD items from the oceanic Rapa Nui region (Table 3) and corresponded to estimated floating times of more than 5 weeks (42.5 days). Along the continental coastline, size and estimated floating times were generally low, with a maximum of two weeks in the southern region, and decreased towards lower latitudes (Table 3). Findings of rafts with larger individuals (8–10 mm capitulum length) were rare in all continental regions, occurring only at the southern continental beaches Mar Brava North and Mar Brava South and the central continental Ritoque beach (Fig. S5). In the northern continental region (Choros beach), no Lepas spp. were found in the sampling section during the entire 12-day sampling period. However, in opportunistic samplings outside the sampling section, we found some samples with such pelagic epibionts. For the pelagic barnacle Jellyella sp., which encrusted high proportions of raft items on Rapa Nui beaches, a quantification of colony diameter was not possible, as they had usually grown together, forming crusts with one or several layers in which individual colonies were no longer recognizable. Nevertheless, it shows that colonies on Rapa Nui litter had established on the litter items at least several weeks before stranding.

Table 3 Maximum capitellum lengths of settled Lepas spp., found on stranded items at each sampling site and estimated minimum floating time.

	Oceanic: Rapa Nui	Continental: South	Continental: Centre	Continental: North	
	Anakena	Ovahe	Mar Brava South	Mar Brava North	Ritoque	Maitencillo	Chorosa	
Lepas sp.	(8)	(1)	(2)	(3)	(28)	(6)	(2)	
Size [mm]	1.9 ± 0.9	3	2.5 ± 0.7	4.5 ± 4.8	2.4 ± 1.6	1.8 ± 0.4	2.3 ± 0.4	
Time [d]	4	6.5	5.4	9.8	5.2	3.9	5.0	
L. anatifera	(2)	(0)	(0)	(0)	(0)	(0)	(0)	
Size [mm]	18.7 ± 4.2	x	x	x	x	x	x	
Time [d]	42.5	x	x	x	x	x	x	
L. pectinata	(0)	(0)	(4)	(5)	(47)	(3)	(3)	
Size [mm]	x	x	4.8 ± 1.7	3.6 ± 1.8	3.0 ± 1.5	3.0 ± 1.0	3.0 ± 1.0	
Time [d]	x	x	13.0	9.7	7.8	8.1	8.1	
L. australis	(0)	(0)	(6)	(4)	(5)	(0)	(0)	
Size [mm]	x	x	5.5 ± 1.7	4.0 ± 1.8	5.0 ± 2.9	x	x	
Time [d]	x	x	12.0	8.7	10.9	x	x	
Notes.

Floating time was calculated based on growth rates reviewed in Thiel & Gutow (2005b). L. anatifera: 0.44 mm day−1; L. pectinata: 0.37 mm day−1; L. australis: 0.46 mm day−1. Highest growth rate (0.46 mm d−1) was assumed for individuals that could not be determined to the species level (Lepas sp.); x, no individuals were identified on samples from this site. Only samples with settled individuals were considered (no larvae). Numbers in parentheses = number of samples considered at each sampling site, pooled from quantitative and opportunistic samplings.

a Only samples from opportunistic samplings; no Lepas spp. found at 1-km section during the sampling period.

Bold values indicate maximum estimated floating time of items for each beach.

Epibionts

More than 50 species from at least nine phyla were identified on the litter items arriving on SE Pacific beaches in the present study (Table S6). Epibiont composition differed between the four sampled regions (Fig. 5). Samples found on oceanic Rapa Nui beaches were dominated by the pelagic bryozoan Jellyella sp., present on ∼90% of all fouled items, and species richness was relatively low (1.2 ± 0.4 and 1.2 ± 0.6 species per raft for Anakena and Ovahe, respectively; Table S6). Pelagic goose barnacles (Lepas sp./L. anatifera) were found on only 7% and 3% of items on Rapa Nui’s Anakena and Ovahe beach, respectively. Some benthic species were present (albeit in low frequencies) on rafts from this region: The coral Pocillopora sp., which is common in the Rapa Nui ecoregion, was present on 3% of fouled items from Anakena beach but was absent on rafts found on Ovahe beach. Spirorbids and other serpulid polychaetes were relatively common on rafts on Ovahe beach (present on 10% of all fouled items), but not on samples from Anakena beach (0.8%).

Figure 5 Frequency of occurrence [%] of the main epibiont taxa amongst all fouled items per beach.

(A) Pelagic epibionts. (B) Benthic epibionts (or habitat unknown). AN, Annellida; AR, Arthropoda; BR, Bryozoa; CN, Cnidaria; MO, Mollusca. SPSG, South Pacific Subtropical Gyre, RN, Rapa Nui; Ana, Anakena; Ova, Ovahe. Conti, Continental; S, South; C, Center; N, North; MBS, Mar Brava South; MBN, Mar Brava North; Rit, Ritoque; Mait, Maitencillo; Cho, Choros.

Rafts in the southern continental region had a slightly richer epibiont community (1.4 ± 0.9 and 1.4 ± 0.8 species per raft on MBS and MBN beach, respectively). Pelagic goose barnacles (Lepas sp., L. pectinata, L. australis,), were more common on MBN than on MBS, being found on 27% and 12% of all fouled items, respectively. Benthic epibionts were frequently encountered on rafts in this region: Benthic encrusting bryozoans were found on approximately 20% of rafts at both MBS and MBN. Mytilid mussels (18% MBS, 7% MBN), polychaetes (12% MBS, 13% MBN) and acorn barnacles (Balanomorpha, 3% MBS, 6% MBN) were also relatively common on rafts in the southern continental region (Fig. 5, Table S6).

An average of 1.7 ± 1.3 and 1.5 ± 0.9 species per raft were found at Ritoque and Maitencillo beach, respectively. Benthic epibionts were more frequent on these beaches than pelagic ones (Lepas spp.; 19% and 13% on Ritoque and Maitencillo, respectively). On rafts on Ritoque beach, the most frequent epibionts were benthic bryozoans with a large diversity (> 10 families), present on 56% of all fouled items on this beach. A single species, Membranipora sp., was found on 36% of fouled items from Ritoque beach. This is most likely M. isabelleana d’Orbigny, 1842 (taxon inquirendum based on WoRMS Editorial Board, 2023), a very common epibiont of local kelp species (e.g., Graiff et al., 2016). Barnacles (orders Balanomorpha and Verrucidae) were frequent on rafts from both central continental beaches, present on 16% and 3% of rafts at both beaches. Identified species from this group were native inhabitants of the local intertidal zone, such as Balanus laevis, Notobalanus flosculus, Austromegabalanus psittacus, Notochthamalus scabrosus and Verruca laevigata. Polychaetes (mainly spirorbids) had colonised 9–10% of rafts at both beaches. Maitencillo beach differed from Ritoque in its high frequency of benthic mussels Mytilidae sp. (41% of all fouled items) and its lower frequency of benthic bryozoans (26% of all fouled items).

The northern continental region (Choros beach) differed from all other regions, due to the absence of pelagic epibionts. In general, species richness was low on rafts at Choros beach (1.1 ± 0.9 species per raft). Mussels (Mytilidae sp.) that are very common in the local intertidal zone and were washed onto the beach in large quantities during the sampling period, were found on almost all (99%) rafts from this beach. Some other benthic epibionts were present in low frequencies: Benthic barnacles (2% of all fouled items), encrusting bryozoans (1.3%) and polychaetes (0.6%; Fig. 5, Table S6).

There were several invasive and cryptogenic species among the epibionts found in our study. Bugula neritina and Bugulina cf. flabellata were found in the northern and central regions, where they are already established. Ciona robusta (Ascidiacea) and Jassa marmorata (Malacostraca), both of which are abundant along the Chilean continental coast and have been found on floating objects before, were found on a buoy stranded on Choros beach. The cryptogenic hydroids Amphisbetia operculata and Plumularia setacea were found entangled with (but not attached to) rafts in the southern and northern continental region, respectively. Several native benthic species were found on AMD rafts for the first time: three species of anemones Anthopleura cf. hermaphroditica, Phymactis papillosa, and Paranthus niveus and four species of intertidal gastropods, Trochita trochiformis, Siphonaria lessoni, Scurria variabilis, Lottia orbignyi, were found on positively buoyant plastic items in the central continental region, within their known range of occurrence. The native amphipod Caprella penantis f. gibbosa, sensu Mayer, a species still awaiting description (Pilar Cabezas et al., 2013 and J Guerra García, pers. comm., 2022) was found on a stranded buoy (Choros beach) and rope (Ritoque beach), constituting the first reported findings of this species on positively buoyant AMD. Another native caprellid, Deutella venenosa, (redescribed in Guerra-García, 2003), which had been reported rafting before (Astudillo et al., 2009), was found on the same stranded rope on Ritoque beach, which constitutes the first record of this species outside the Coquimbo region.

Discussion

AMD accumulation rates and composition

Overall, AMD accumulation rates were not significantly different between the oceanic Rapa Nui region and the continental regions. Average accumulation rates on the Rapa Nui beaches Ovahe and Anakena (300 and 200 items km−1 day−1) were similar to those measured on the Apina Nui pocket beach of the same island in a study by Thiel et al. (2021) with a similar sampling protocol (230 items km−1 day−1), indicating that the 10-day sampling period allowed obtaining robust average values. Maximum daily accumulation rate, however, was much higher in the previous study (1,900 items km−1 day−1; Thiel et al., 2021) than in the present one, possibly due to the longer sampling period of 190 days, which included a wider range of climatic conditions (e.g., strong winds). Along the continental coast, highest AMD arrival frequencies (both with and without pelagic epibionts) and the prevalence of typical locally-sourced litter items in the densely populated central region, where the beaches Ritoque and Maitencillo are located downstream of several major cities and economic/industrial hotspots, confirm that population density is in fact an important predictor of regional (short distance) litter accumulation along the SE Pacific continental coast, as suggested by Chenillat et al. (2021).

Ours is the first study reporting daily accumulation rates for Chilean continental beaches, and comparisons with studies from other global sites are difficult due to differences in sampling methodology. However, high rates of daily AMD accumulation in regions with high population densities and recreational beach use were also found in studies from the South African Cape Town area, and the Australian and Brazilian east coasts, highlighting the impact of local inputs of single-use items and other consumer plastics (Santos et al., 2005; Smith & Markic, 2013; Ryan et al., 2014; Chitaka & Von Blottnitz, 2019). There were also interesting differences between beaches in the same regions or even neighbouring ones, with total AMD pollution (with and without epibionts) being much higher on Ritoque than on Maitencillo beach (central continental region) and much higher in MBS than on MBN (southern continental region). In the central region, Ritoque beach lies in close proximity to the Aconcagua River and is more exposed than Maitencillo, which could influence litter deposition from local sources, as was shown for several rivers along the Chilean continental coast in a previous study (Rech et al., 2014). In the southern region, MBS is also influenced by the mouth of a small river, possibly explaining the higher frequency of AMD there. There are several factors (and their interactions) that are thought to influence litter deposition within a beach or region, like orientation, exposure, presence of protecting structures, and complexity of the foreshore, as well as local (small-scale) circulation and currents (see for example Critchell & Lambrechts, 2016).

AMD accumulating along the continental coast was mostly of local origin, with high proportions of non-durable items in the central region (particularly food wrappers and other thin foils, which fragment and/or sink quickly) and high proportions of ropes in the southern and northern regions, where population density is low and fishing is the predominant economic activity. While no past studies with comparable litter categories are available for the northern and central continental regions, the high proportions of fishing- and aquaculture-related litter in the southern continental region are in line with the results of previous surveys by Perez-Venegas et al. (2017) and Ahrendt et al. (2021). Litter composition found in our study along the continental coastline reflects a global pattern, with food packaging and items from fishing activities as the main components (e.g., Chitaka & Von Blottnitz, 2019; Hardesty et al., 2021; Morales-Caselles et al., 2021). On the contrary, and in agreement with prior studies (Rech et al., 2018; Thiel et al., 2021), AMD accumulating at Rapa Nui shores in the centre of the SPSG comprised primarily durable hard plastic items that support the presence of considerable fouling weight without sinking, as well as ropes. Such items reach the SPSG centre region mainly from the SE Pacific Chilean continental coast and its offshore fishing grounds (Van Gennip et al., 2019).

AMD with pelagic epibionts and floating times

Accumulation rates of rafts with a pelagic trajectory, as indicated by the presence of pelagic epibionts, were significantly higher on oceanic Rapa Nui beaches than in all other regions, which is in line with our hypothesis. Along the continental coast, highest frequencies of objects with pelagic epibionts were expected in the southern region, where the West Wind Drift meets the South American continent. However, in the present study, arrival rates of rafts with pelagic epibionts (Lepas spp.) were higher in the central continental region. Maximum capitellum lengths of Lepas spp., on the other hand, pointed to longer floating times of rafts stranding in the southern region. Estimated floating times of AMD rafts in all three continental regions during our sampling season (autumn) were in the same range as those of floating kelp studied by López et al. (2017) in the same regions in winter. The latter, as well as other previous studies on floating kelp, showed that currents and winds influence the transport and stranding dynamics of floating objects along the Chilean continental coast. The prevailing upwelling regime directs floating items offshore, where they may accumulate in retention zones formed by stable wind and current regimes. During events of upwelling relaxation or during strong onshore winds, such items might be pushed onto the coast (Hinojosa, Rivadeneira & Thiel, 2011; López et al., 2017; Schreiber et al., 2020).

Recent studies confirm seasonal and intra-seasonal variability of surface currents and nearshore circulation in both the central and southern continental regions (Letelier, Pizarro & Nuñez, 2009; Strub et al., 2019; Aguirre et al., 2021), with upwelling maxima in austral summer and short-term (3–7 days) fluctuations throughout the year (Letelier, Pizarro & Nuñez, 2009). Given that our study was carried out during autumn, when upwelling is less dominant in the central region and during a period of local storm surges, it can be assumed that these conditions caused the relatively high frequencies of rafts with pelagic epibionts found there. This is corroborated by the relatively large proportion of non-durable items (e.g., food wrappers and other packaging) among rafts with pelagic epibionts in this region.

In the oceanic Rapa Nui region, with estimated floating times of more than 6 weeks, almost all raft items were durable hard plastic items or fragments. Patterns of offshore accumulation of floating items and subsequent stranding due to wind and tides has also been suggested for Rapa Nui and the Chilean fjord regions (Hinojosa et al., 2010; Hinojosa, Rivadeneira & Thiel, 2011; Thiel et al., 2021), giving a likely explanation for the large daily variations in AMD stranding rates observed on all sampled beaches in our study. This corroborates the importance of small-scale (in time and space) meteorological events, particularly storms, as pointed out by other authors (e.g., Waters & Craw, 2018). While the present study used 12 consecutive sampling days (10 days on Rapa Nui beaches) and deduced latitudinal differences in pelagic litter arrivals primarily on the basis of floating times calculated from epibiont size, it is probable that the use of repeated or prolonged sampling events, covering a wider range of climatic, and particularly, wind-related conditions (see Thiel et al., 2021), would result in a clearer latitudinal pattern. Nevertheless, despite the rather short sampling period, our study covered a relatively wide variation of AMD arrival frequencies on the sampled beaches and provides a first quantitative comparison on a large latitudinal scale.

Epibiotic community/benthic species

Scientific reports of coastal invertebrates rafting on artificial objects go back several decades (see review in Kiessling, Gutow & Thiel, 2015) and on natural objects even further (reviewed in Thiel & Gutow, 2005b). Floating AMD is increasingly considered an important dispersal vector for invasive benthic/intertidal species, with numerous examples (e.g., Carlton et al., 2017; Clarke Murray et al., 2019; García-Gómez, Garrigós & Garrigós, 2021; Haram et al., 2021; Haram et al., 2023). Corals of the genus Pocillopora, which we found attached to plastics stranded on Rapa Nui, have long been suggested to disperse by rafting on floating material, with several records of long-distance (tens of thousands of kms and several years) voyages (e.g., Jokiel, 1984; Jokiel, 1989; Jokiel, 1990; Carlton et al., 2017 in the North Pacific; Bryan et al., 2004 in the South Pacific). We have also found Pocillopora sp. colonizing AMD rafts in previous studies in the SPSG’s Rapa Nui region (Rech et al., 2018; Rech et al., 2021), confirming the relatively high frequency of coral rafting. Several species of the same order (Scleractinia), amongst them invasive Tubastraea spp., are also known from floating AMD (Hoeksema, Roos & Cadée, 2015; Hoeksema, Pedoja & Poprawski, 2018; Mantelatto et al., 2020; Soares et al., 2022).

Sea anemones (order Actiniaria) are also frequently found rafting, which is thought to facilitate invasions (Astudillo et al., 2009; Goldstein, Carson & Eriksen, 2014; Glon et al., 2020). Anthopleura spp. occurred with particularly high frequencies on floating plastics from the North Pacific Subtropical Gyre and are considered members of the “neopelagic” community (Haram et al., 2021; Haram et al., 2023). The amphipod Caprella penantis cf. gibbosa, sensu Mayer is known from floating buoys in the sampling region (Astudillo et al., 2009), which may have facilitated a recent northward range extension (Chunga-Llauce & Pacheco, 2021). The relatively high diversity and frequency of members of benthic, intertidal fouling communities (>50% of all fouled items on Ritoque beach) points to the growing possibility of dispersal and range expansion through AMD rafting for native invertebrates.

Based on our results, the Chilean continental coast could be pictured as an ideal source region of a diverse array of AMD-rafting benthic invertebrates, which are diverted offshore by upwelling regimes and then relatively quickly transported to the SPSG’s central region (see Van Gennip et al., 2019). This is in line with a study by Astudillo et al. (2009), who found a rich and diverse invertebrate community on detached aquaculture buoys off the continental Coquimbo region. Nevertheless, in accordance with a more local, previous study (Rech et al., 2018), we did not find any non-indigenous benthic epibionts on rafts on Rapa Nui beaches, which were usually encrusted only by the pelagic bryozoan Jellyella sp., or (in few cases) carried local (native) benthic species. The discrepancy between a suspected high frequency of rafting invertebrates’ departure from the continental coast and their apparent lack of arrival in the Rapa Nui region is in line with a previous study on floating objects (Rech et al., 2021) and is thought to be due to the SPSG’s hyperoligotrophy, in combination with the extreme nutrient and temperature gradient between the alongshore Humboldt current system and the open ocean that may limit continental epibionts’ survival. Another factor may be the harshness of landing on (or, rather, being thrown against) Rapa Nui’s mostly rocky shores that would probably tear most larger epibionts off their rafts. This is also a likely explanation for the relatively low frequency and body size of pelagic goose barnacles Lepas spp. found on rafts stranded there, as opposed to rafts floating off Rapa Nui sampled in Rech et al. (2021). Furthermore, despite its location in the very centre of the SPSG, the island of Rapa Nui is very small (∼20 km in length) in comparison to the vast oceanic region in which it resides, drastically limiting the probability of a given raft or epibiont landing on its shores. Given that there are no other, larger stretches of land that could be monitored, we may assume that most rafts and rafters will go undetected in this region.

Several characteristics (hyperoligotrophy, large extension, general lack of monitoring) differentiate the SPSG from other ocean gyres, like the relatively small North Atlantic Subtropical Gyre, where benthic rafters from East coast are frequently found on long and easily accessible sandy beaches of the Western European continental coast (e.g., Hoeksema, Roos & Cadée, 2012; Hoeksema, Roos & Cadée, 2015; Hoeksema, Pedoja & Poprawski, 2018; Holmes et al., 2015). While, due to the reasons detailed above, we cannot draw definite conclusions about the frequencies and possible impact of continental rafting biota in the SPSG centre region, our results do indicate that rafting may play a role in local benthic species’ dispersal among the oceanic islands within or near the gyre (e.g., coral Pocillopora sp., see above), as has been recently discussed for the chthamalid barnacle Rehderella belyaevi, an abundant member of Rapa Nui’s intertidal communities (Wares et al., 2022).

Conclusions

The present study is the first to quantify arrival frequencies of floating AMD, with and without pelagic epibionts, on a daily basis, at several strategic locations covering an important part of the SE Pacific coasts. Our results corroborate the predictions made by recent models based on the main oceanic wind and surface currents, showing that those are reliable and cost-efficient tools for identification of high-risk arrival sites of AMD-rafting biota on large geographic scales. The regions mostly affected by AMD from distant sources are, as hypothesized, the South Pacific Subtropical Gyre’s centre region and, to a much lesser extent, the southern continental coast (based on estimated floating times). High variation in daily arrival frequencies of rafts with pelagic trajectories on smaller geographic scales (i.e., at a sampling site or between neighbouring sites with a distance of <50 km) is in accordance with the results of previous studies on kelp and AMD rafting in the SE Pacific and confirms the suggested influence of temporary and seasonal climatic conditions (fluctuations in upwelling regimes, storm surges), as well as coastline geomorphology. A diverse pool of benthic AMD rafters was found along the SE Pacific continental coast, which is thought to facilitate alongshore dispersal or range expansions. Currently, these continental rafters do not seem to arrive on Rapa Nui shores in significant frequencies, possibly due to unfavourable conditions of both the SPSG’s waters and the island’s geomorphology. Nevertheless, the recurrent findings of local benthic coral Pocillopora sp., with a known history of rafting, on Rapa Nui rafts, corroborates the possible importance of rafting for benthic species dispersal among South Pacific oceanic islands. Apart from their implications for NIS dispersal and marine biogeography, our results show once more how consumer plastics, and particularly single-use items, contaminate and spread along whole oceanic systems and reinforce the urgent need to drastically reduce production of such items.

Supplemental Information

Table S1 Buoyancy of items by region

Buoyancy was tested by submerging each item in sea water (for details see Material & Methods). N, number of items. n.d., not determined. Items from quantitative and opportunistic samplings are considered.

Click here for additional data file.

Table S2 Buoyancy by category

n, number of items tested in each category. Buoyancy: positive (item floats), negative (item sinks), or n.d., not determined (not tested or result unclear).

Click here for additional data file.

Table S3 Arrival rates of litter items [km−1] by sampling day, as well as mean ± standard deviation (∅) per beach and region

(a) For all AMD, (b) for items with pelagic epibionts only.

Click here for additional data file.

Table S4 Detailed composition of AMD found during quantitative daily samplings within the defined sampling transect of each beach

All, all items; pel, items with pelagic epibionts only. N, number of items considered. Main categories (bold) and subcategories with a minimum frequency of 5% at any sampling site are shown. Subcategories with a frequency of <5% were pooled. * no items with pelagic epibionts were found during quantitative daily samplings on Choros beach.

Click here for additional data file.

Table S5 Results of similarity percentage analysis (SIMPER), based on Bray-Curtis similarities, showing within-group similarities in AMD composition for the four sampled regions

(A) All AMD items, (B) only items with pelagic epibionts. Only items found during quantitative daily samplings within the defined sampling sections were included (no items from opportunistic samplings).

Click here for additional data file.

Table S6 Species list

Frequency of occurrence of each taxon amongst all fouled items. yes, found in opportunistic sampling only (=not found during quantitative samplings). (yes) = entangled, but not attached. n.i. = not (further) identified. Habitat: ben = benthic, pel = pelagic. # = found on negatively buoyant items only. ∑taxa per beach = minimum number of distinguishable taxa per sampled beach.

Click here for additional data file.

Figure S1 Deposition of subtidal anthropogenic litter, red algae and invertebrates on sampling transect of Choros beach

Click here for additional data file.

Figure S2 Schematic overview of beach sections

Red: “Recent” intertidal with tidelines from the last 24 hours (=sampled beach section). Blue: “Ancient” intertidal, with tidelines older than 24 hours (not sampled). Yellow: Dunes (not sampled).

Click here for additional data file.

Figure S3 Daily arrival rates [items * km−1] of AMD (all items) in the defined sampling section on each sampled beach for each sampling day

SPSG, South Pacific Subtropical Gyre; Conti, Continental.

Click here for additional data file.

Figure S4 Daily arrival rates [items * km−1] of litter items with pelagic epibionts in the defined sampling section on each sampled beach for each sampling day

SPSG, South Pacific Subtropical Gyre; Conti, Continental. No litter items carrying pelagic epibionts were found during quantitative daily samplings on Choros beach in the northern continental region.

Click here for additional data file.

Figure S5 Maximum capitellum lengths of Lepas spp. at sampled beaches

Samples from quantitative and opportunistic samplings are pooled. n, number of individuals considered at each beach. Boxplot shows average, median, quartiles and outliers for each site, based on the largest individual of each identifiable species of each raft. SPSG, South Pacific Subtropical Gyre. Conti, Continental; MBS, Mar Brava South; MBN, Mar Brava North.

Click here for additional data file.

Figure S6 Examples of “rope bundles” with entangled biota from Mar Brava South

White bundle on the left is a fragment of a woven plastic bag, rather than a rope bundle.

Click here for additional data file.

Text S1 Supplemental text (Material & Methods)

Additional information about sampled sites and regions, as well as the sampled beach sections.

Click here for additional data file.

Data S1 Raw data

Daily litter accumulation frequencies (all AMD and AMD with pelagic epibionts only) by item category and beach, epibiont matrices including all rafts on all beaches, results of buoyancy experiments and measurements of Lepas spp. capitellum lengths.

Click here for additional data file.

We are very grateful to the taxonomists who kindly helped with species identification: Jan Beermann (Amphipoda), James T. Carlton (Balanomorpha and miscellaneous invertebrates), Kenneth Finger (Foraminifera), José Guerra-García (Caprellidae), Bert Hoeksema (corals), Horia R. Galea (Hydrozoa), Leslie Harris (Polychaeta), Vreni Häusermann and Carlos Spano (Actiniaria). We sincerely thank Dr. Jan Hafner for his help with Fig. 1 and Stephen Sampson for revision of the English language.

Additional Information and Declarations

Competing Interests

Author Contributions

Field Study Permissions

Data Availability

The authors declare there are no competing interests.

Sabine Rech conceived and designed the study, performed the study, analyzed the data, prepared figures and/or tables, authored or reviewed drafts of the article, and approved the final draft.

Rene Matias Arias performed the study, authored or reviewed drafts of the article, and approved the final draft.

Simón Vadell performed the study, authored or reviewed drafts of the article, and approved the final draft.

Dennis Gordon analyzed the data, authored or reviewed drafts of the article, and approved the final draft.

Martin Thiel conceived and designed the study, authored or reviewed drafts of the article, and approved the final draft.

The following information was supplied relating to field study approvals (i.e., approving body and any reference numbers):

All field work was approved by the Ethics Comittee (Comité Ético Científico) of Universidad Católica del Norte.

The following information was supplied regarding data availability:

The raw data is available in the Supplemental File.

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
