# Peer review of "Daily accumulation rates of floating debris and attached biota on continental and oceanic island shores in the SE Pacific: testing predictions based on global models"

_PeerJ, doi:10.7717/peerj.15550_

## Round 0.1 · original submission · Major Revisions

Please incorporate all recommendations of the reviewers and re-submit with point-to-point answers to the comments. Please put more clarity to the sample of the study.

·

Basic reporting

This is an important paper and highlights the lack of monitoring of rafting biota on shores worldwide and discusses solutions to maximise the use of people’s time by using predictive modelling to estimate the most likely areas of plastic accumulation and hence rafting biota buildup. It is clear that these results are relevant to the South Pacific Subtropical Gyre but the model can be applied to anywhere in the world.
The use of English is excellent, with very little to correct and it is completely unambiguous. References used are suitable for the subject matter and are used in the correct context. Figures and table are referred to correctly and the paper stands alone without need for additional reading.
A few very minor changes are suggested:
Abstract line 25 : ‘important’ not quite the right word here, ‘Significant’ perhaps or needs another word to qualify it. In what way is it an important impact? Positive, negative?
Line 214 ‘marine ecology laboratory’ where? University? State the name and location.
Line 223 ‘taxonomist’ should be ‘taxonomists’ plural
Line 247 ‘stranding’ Do you mean that you wanted to test the buoyancy of the item upon stranding or when it was stranded? Change this line depending on the which of the above.
Line 261 ‘in’ should be ‘into’
Line 214 ‘marine ecology laboratory’ – Which lab? Facultad de Ciencias del mar?

Experimental design

Clearly this subject has been very well-researched, and the experiment carried out in an organized and methodical manner.
This manuscript adheres to PeerJ’s Aims and Scope of the journal. It is clear what the research question was and how the authors set about creating the experiment to answer it. This piece of work undoubtedly fills in a knowledge gap. I am aware of a worldwide project run by some of the authors to further determine routes of rafts and epibionts. This sort of work may encourage others in geographical areas where there are gaps in the rafting data to carry out the same work.

Validity of the findings

The results are interesting and the explanations in the discussion are well reasoned. The Conclusions are articulate and are certainly linked to the original research question.

Additional comments

I see no problems with this paper and would recommend that it is published with minor changes.

Reviewer 2 ·

Basic reporting

The manuscript is constructed around well-thought hypothesis with a main goal of test if the existing models can predict the arrival areas of rafting taxa on marine debris. The introduction guides the reader through the theoretical background needed to understand the hypothesis. In the current form the connection between the results and the hypothesis is not clear. There are some statements in the discussion that cannot be easily related to the results, such as lines 421-423, 428-430 and 434-438. I am not saying that the results presented in the manuscript can’t be used to draw such conclusions, but authors need to clearly communicate how their findings (and methods) led to such conclusions. I think that the manuscript would benefit if the methods and results were constructed around the hypothesis.

Experimental design

The diversity of sampled areas is a strength of this study. However, I think that some parts of the methods need more detailed information. Below are some questions that I think, when answered, will improve the manuscript's quality.
Are there beach cleaning efforts in any of the study areas? If yes, how will this impact the findings?
In the methods, please specify when the beaches were sampled.
It authors should provide more information on how the sampling efforts where done, especially regarding the area sampled in the beach. It would be great if a schematic figure was present in the manuscript or if some standard protocol was cited as a reference to detailed information about the methods.
It would be great if the classification of the items followed existing protocols. I suggest https://doi.org/10.1038/s41893-021-00720-8 as a reference.
Line 245: Please define what was considered several days.
Please provide more information on how the analyses reported on lines 269-271 were performed.
It would be appreciated if authors find ways to present the results about the epibionts in a graphical way. For example, the frequency of occurrence of each taxon found on the beaches.

Validity of the findings

In line with my comments in the previous section, I have some additional questions that I think should be clarified to improve confidence in the results and the relationship between the results and the conclusion of the manuscript.

Why seven beaches are enough to test your hypothesis, considering that the selected beaches vary greatly in their characteristics?
How do the differences in the size of sampling sections among beaches impact the results?
How do the slight differences in the methods for each area (lines: 184, 198-203; 247-249) impact the results?
Why twelve days are enough to test your hypothesis?
Line 421-423: I am not sure if you can backup this statement with the methods and results presented in the manuscript. If you think you that you can state this, please, walk the readers through your methods and results to make this point in more clear way.

Line 428-430: I think that if the characteristics of the study areas are important to draw some conclusions, the characterization of the study should be improved and structured in a way that guides the readers.

Lines 434: It is not clear for me how an anthropogenic debris were classified as a local in its origin. For this topic, I recommend providing more details on the methods.

Lines 434-437: Same comments from lines 428-430.

Lines 538-540: In the current structure, I am not sure if this statement can be fully backed up by the methods used and the results presented. However, it may be clarified by addressing some of my previous comments.

---

## Round 0.2 · Minor Revisions

Congratulations. The paper has been improved after revisions and may be accepted for publication. However, the language of the paper need revisions. Please take the help of fluent English-speaking colleagues to improve the language.

·

Basic reporting

The authors have completed the minor changes suggested and I think it is ready for publication.

Experimental design

No comment

Validity of the findings

No comment

Additional comments

This is ready for publication

Reviewer 3 ·

Basic reporting

The paper is written nicely. A minor revision is suggested in the Abstract, Results and Discussion and Conclusion
Abstract: Direct results are written. It is better to write 2-3 lines, why this study is important to conduct. What methods/material you used for this study.
At the end of Abstract, write a line what is new in your study or what are implications of your results.
Introduction Accept my appreciation for writing good introduction. However, write a paragraph regarding statement of problem at the end of introduction.

Experimental design

Daily survey effectively conducted. Overall, material and methods clearly understood. Well-done.

Validity of the findings

Results are truly explained however, that needed to justified. Some results are justified in discussion section however, all results needed to justify based on past literature

Additional comments

You needed to write your own finding based on results obtained by comparing/contrasting past studies. What is new in your results. Just write those in conclusion.

---

## Round 0.3 · accepted · Accept

The language of the paper is improved, and the paper is accepted for publication.